# Graph Neural Modeling of Network Flows

## Abstract

Network flow problems, which involve distributing traffic such that the underlying infrastructure is used effectively, are ubiquitous in transportation and logistics. Among them, the general Multi-Commodity Network Flow (MCNF) problem concerns the distribution of multiple flows of different sizes between several sources and sinks, while achieving effective utilization of the links. Due to the appeal of data-driven optimization, these problems have increasingly been approached using graph learning methods. In this paper, we propose a novel graph learning architecture for network flow problems called Per-Edge Weights (PEW). This method builds on a Graph Attention Network and uses distinctly parametrized message functions along each link. We extensively evaluate the proposed solution through an Internet flow routing case study using 21 Service Provider topologies and 2 routing schemes. We show that, with both synthetic and real-world traffic, PEW yields substantial gains over architectures whose global message function constrains the routing unnecessarily. We also find that an MLP is competitive with other standard architectures. Furthermore, we analyze the relationship between graph structure and predictive performance for data-driven routing of flows, an aspect that has not been considered by existing work in the area.

## 1 Introduction

Flow routing represents a fundamental problem that captures a variety of optimization scenarios that arise in real-world networks (Ahuja, 1993, Chapter 17). The Multi-Commodity Network Flow problem allows for multiple flows of different sizes between several sources and sinks that share the same distribution network. It is able to formalize the distribution of goods in a logistics network, of cars in a rail network (Hu, 1963) and, most relevantly to the present paper, of packets in a computer network. We illustrate MCNF problems in Figure 1.

Many works in recent years have considered data-driven approaches for computer networking (Jiang et al., 2017) and self-driving networks (Feamster & Rexford, 2018). In this area, machine learning models have been leveraged for (1) predicting the behavior of an existing routing scheme in unseen circumstances (Geyer & Carle, 2018; Rusek et al., 2019) and (2) learning the routing scheme itself (Valadarsky et al., 2017; Almasan et al., 2021). Early works in this area were based on MLP architectures (Valadarsky et al., 2017; Reis et al., 2019). More recently, models purposely designed to operate on graphs, including variants of the expressive Message Passing Neural Networks (Rusek et al., 2019; Almasan et al., 2021) and Graph Nets (Battaglia et al., 2018), have been adopted.

Learning representations are an important underlying aspect of this wave of works. Developing an effective architecture is hence fundamental to the application of machine learning in flow routing scenarios, which is the task we set out to address in this paper. Despite the promise of graph learning, current works nevertheless adopt architectures that aggregate messages along neighboring edges using the same message functions. In the context of routing flows, this constrains the model unnecessarily. Instead, we argue that nodes should be able to weight flows along each link separately, so that each node may independently update its state given incoming and outgoing traffic, leading to better *algorithmic alignment* (Xu et al., 2020) between the computational mechanism of the GNN and the task. We illustrate this in Figure 2.

Furthermore, the ways in which prior works encode the demands as node features varies between the full demand matrix (Valadarsky et al., 2017; Zhang et al., 2020) and a node-wise summation (Hope & Yoneki,

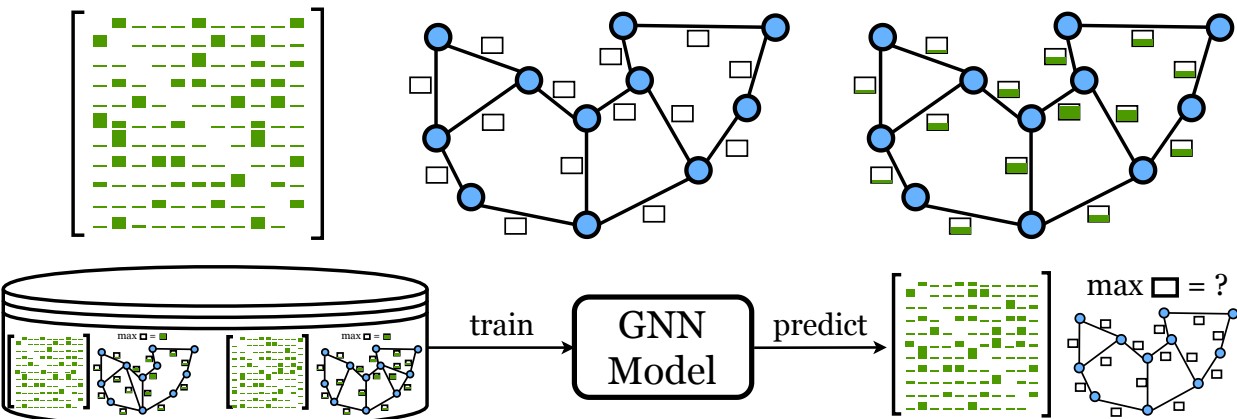

Figure 1: **Top.** An illustration of the Multi-Commodity Network Flow family of problems. The requirements of the routing problem are defined using a graph topology in which links are equipped with capacities and a matrix that specifies the total amount of traffic that has to be routed between each pair of nodes. All flows have an entry and exit node and share the same underlying transportation infrastructure. Under a particular routing scheme, such as shortest path routing, the load of each link is set to the total amount of traffic passing over it. **Bottom.** A model is trained using a dataset of the Maximum Link Utilizations for certain demand matrices and graph topologies. It is then used to predict the Maximum Link Utilization for an unseen demand matrix.

2021), and it is unclear when either is beneficial. Besides the learning representation aspects, existing approaches in this area are trained using very few graph topologies (typically 1 or 2) of small sizes (typically below 20 nodes). This makes it difficult to assess the gain that graph learning solutions bring over vanilla architectures such as the MLP. Additionally, a critical point that has not been considered is the impact of the underlying graph topology on the effectiveness of the learning process. To address these shortcomings, we make contributions along the following axes:

- **Learning representations for data-driven flow routing.** We propose a novel mechanism for aggregating messages along each link with a different parametrization, which we refer to as *Per-Edge Weights (PEW)*. We propose an instantiation that extends the GAT (Veličković et al., 2018) via a construction akin to the RGAT (Busbridge et al., 2019). Despite its simplicity, we show that this mechanism yields substantial predictive gains over architectures that use the same message function for all neighbors. We also find that PEW can exploit the complete demand matrix as node features, while the GAT performs better with the lossy node-wise sum used in prior work.

- **Rigorous and systematic evaluation.** Whereas existing works test on few, small-scale topologies, we evaluate the proposed method and 4 baselines on 21 real-world Internet Service Provider topologies and 2 routing schemes in the context of a case study in computer networks, yielding 86400 independent model training runs. We evaluate learning architectures' ability to predict the Maximum Link Utilization under a given routing scheme when faced with unseen demands generated by a synthetic model as well as real-world traffic measurements, corresponding to scenario (1) described above. Perhaps surprisingly, we find that a well-tuned MLP is competitive with other GNN architectures when given an equal hyperparameter and training budget.

- **Understanding the impact of topology.** The range of experiments we carry out allows us to establish that a strong link exists between topology and the difficulty of the prediction task, which is consistent across routing schemes and learning architectures. Generally, predictive performance decreases with graph size and increases with heterogeneity in the local node and edge properties. Moreover, we find that, when graph structure varies through the presence of different subsets of nodes, the performance of GNNs increases compared to structure-agnostic methods, such as MLP.

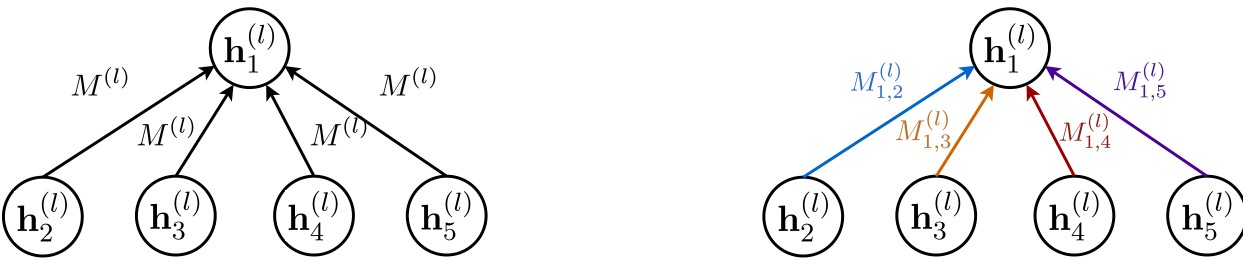

Figure 2: **Left.** An illustration of the MPNN used in previous flow routing works, which uses the same message function $M^{(l)}$ for aggregating neighbor messages. **Right.** An illustration of our proposed Per-Edge Weights (PEW), which uses uniquely parametrized per-edge message functions.

## 2 Related work

**Flow problems and solution methods.** One classic example is the maximum flow problem, which seeks to find the best (in terms of maximum capacity) path between a source node and a sink node. For maximum flow problems, efficient algorithms have been developed (Cormen et al., 2022, Chapter 26), including a recent near-linear time approach (Chen et al., 2022). For the more complex MCNF problems, Linear Programming solutions can be leveraged in order to compute, in polynomial time, the optimal routes given knowledge of pairwise demands between the nodes in the graph (Fortz & Thorup, 2000; Tardos, 1986). At the other end of the spectrum, oblivious routing methods derive routing strategies with partial or no knowledge of traffic demands, optimizing for "worst-case" performance (Räcke, 2008).

As recognized by existing works, *a priori* knowledge of the full demand matrix is an unrealistic assumption, as loads in real systems continuously change (Feldmann et al., 2001). Instead, ML techniques may enable a middle ground (Valadarsky et al., 2017): learning a model trained on past loads that can perform well in a variety of traffic scenarios, without requiring a disruptive redeployment of the routing strategy (Fortz & Thorup, 2002).

**Neural networks operating on graphs.** Much effort has been devoted in recent years to developing neural network architectures operating on graphs. Several approaches, such as Graph Convolutional Networks (GCNs), use convolutional filters based on the graph spectrum, which can be approximated efficiently (Defferrard et al., 2016; Kipf & Welling, 2017). An alternative line of work is based on message passing on graphs (Sperduti & Starita, 1997; Scarselli et al., 2009) as a means of deriving vectorial embeddings. Both Message Passing Neural Networks (MPNNs) (Gilmer et al., 2017) and Graph Networks (Battaglia et al., 2018) are attempts to unify related methods in this space, abstracting the commonalities of existing approaches with a set of primitive functions.

Expressivity is another major concern in the design of this class of architectures. Notably, Gated Graph Neural Networks (GG-NNs) (Li et al., 2017) add gating mechanisms and support for different relation types, as well as removing the need to run the message propagation to convergence. Graph Attention Networks (GATs) (Veličković et al., 2018) propose the use of attention mechanisms as a way to perform flexible aggregation of neighbor features. Relational learning models for knowledge graphs, such as the RGCNs (Schlichtkrull et al., 2018) that extends the GCN architecture, use different parametrizations for edges with different types. The RGATs (Busbridge et al., 2019) follow the blueprint of RGCNs and extend the GAT approach to the relational setting. Despite the tremendous success of relational models for a variety of tasks, perhaps surprisingly, recent work shows that randomly trained relation weights may perform similarly well (Degraeve et al., 2022).

**ML for routing flows in computer networks.** Several works have considered machine learning approaches to perform supervised learning for routing flows in computer networks. (Geyer & Carle, 2018) proposes a variant of the GG-NN and trains it to predict paths taken by conventional routing algorithms.

The work of (Rusek et al., 2019) proposes a MPNN variant and uses it to predict graph-level metrics such as delay and jitter. (Reis et al., 2019) uses an MLP representation and supervised learning to predict the full path that a flow should take through the network.

Other works have considered reinforcement learning the routing protocol itself in a variety of problem formulations: (Valadarsky et al., 2017) uses an MLP and considers learning per-edge coefficients that are used with "softmin" routing. (Xu et al., 2018) proposes an MLP approach for learning traffic split ratios for a set of candidate paths. (Zhang et al., 2020) uses a CNN to re-route a proportion of important (critical) flows. (Almasan et al., 2021) proposes a formulation that routes flows sequentially, which then become part of the state. It uses a MPNN representation. Most recently, (Hope & Yoneki, 2021) adopts the formulation in (Valadarsky et al., 2017), showing that the use of Graph Networks improves performance when applied to one graph topology.

**Algorithmic reasoning and combinatorial optimization.** Another relevant area is algorithmic reasoning (Veličković et al., 2018; Cappart et al., 2021), which trains neural networks to execute the steps taken by classic algorithms (Cormen et al., 2022) with the goal of obtaining strong generalization on larger unseen inputs. (Georgiev & Liò, 2020) trains a MPNN to mimic the steps taken by the Ford-Fulkerson maximum flow algorithm (Cormen et al., 2022, Chapter 24). An important difference to this line of work is that in our case our model does not include any knowledge of the routing scheme, while the approaches based on algorithmic reasoning use the granular algorithm steps themselves as the supervision signal. More broadly, our work belongs to a recent wave of interest in using graph neural networks for combinatorial optimization problems over graphs, in which the goal is to use machine-learned components to replace or assist traditional solution methods (Khalil et al., 2017; Bengio et al., 2021; Schuetz et al., 2022; Gamarnik, 2023).

## 3 Methods

### 3.1 Routing formalization and learning task

**Flow routing formalization.** We assume the splittable-flow routing formalization proposed by Fortz & Thorup (2004). We let $G = (V, E)$ be a directed graph, with $V$ representing the set of nodes and $E$ the set of edges. We use $N = |V|$ and $m = |E|$ as shorthands, as well as $v_i$ and $e_{i,j}$ to denote specific nodes and edges, respectively. Each edge has an associated *capacity* $\kappa(e_{i,j}) \in \mathbb{R}^+$. We also define a *demand matrix* $\mathbf{D} \in \mathbb{R}^{N \times N}$ where entry $D_{src,dst}$ is the traffic that source node *src* sends to destination *dst*. With each tuple $(src, dst, e_{i,j}) \in V \times V \times E$ we associate the quantity $f_{e_{i,j}}^{(src,dst)} \geq 0$, which specifies the amount of traffic flow from *src* to *dst* that goes over the edge $e_{i,j}$. The *load* of edge $e_{i,j}$, $\text{load}(e_{i,j})$, is the total traffic flow traversing it, i.e., $\text{load}(e_{i,j}) = \sum_{(src,dst) \in V \times V} f_{e_{i,j}}^{(src,dst)}$. Furthermore, the quantities $f_{e_{i,j}}^{(src,dst)}$ must obey the following flow conservation constraints:

$$\sum_{e \in \delta^+(v_i)} f_e^{(src,dst)} - \sum_{e \in \delta^-(v_i)} f_e^{(src,dst)} = \begin{cases} D_{src,dst} & \text{if } v_i = src, \\ -D_{src,dst} & \text{if } v_i = dst, \\ 0 & \text{otherwise.} \end{cases} \tag{1}$$

where the sets $\delta^+(v_i), \delta^-(v_i)$ are node $v_i$'s outgoing and incoming edges, respectively. Intuitively, these constraints capture the fact that traffic sent from *src* to *dst* originates at the source (first clause), must be absorbed at the target (second clause), and ingress equals egress for all other nodes (final clause).

**Routing schemes.** A routing scheme $\mathscr{R}$ specifies how to distribute the traffic flows. Specifically, we consider two well-known routing schemes. The first is the *Standard Shortest Paths* (SSP) scheme in which, for a given node, the full flow quantity with destination *dst* is sent to the neighbor on the shortest path to *dst*. The widely used *ECMP* scheme (Hopps, 2000) instead splits outgoing traffic among all the neighbors on the shortest path to *dst* if multiple such neighbors exist.

**Prediction target.** A common way of evaluating a routing strategy $\mathscr{R}$ is *Maximum Link Utilization (MLU)*, i.e., the maximal ratio between link load and capacity. Formally, given a demand matrix $\mathbf{D}$, we denote it as

$\text{MLU}(\mathbf{D}) = \max_{e_{i,j} \in E} \frac{\text{load}(e_{i,j})}{\kappa(e_{i,j})}$. This target metric has been extensively studied in prior work (Kandula et al., 2005) and is often used by ISPs to gauge when the underlying infrastructure needs to be upgraded (Guichard et al., 2005).

**Supervised learning setup.** We assume that we are provided with a dataset of traffic matrices $\mathcal{D} = \cup_k \{\mathbf{D}^{(k)}, \text{MLU}(\mathbf{D}^{(k)})\}$. Given that our model produces an approximation $\widehat{\text{MLU}}(\mathbf{D}^{(k)})$ of the true Maximum Link Utilization, the goal is to minimize the Mean Squared Error $\frac{\sum_k (\text{MLU}(\mathbf{D}^{(k)}) - \widehat{\text{MLU}}(\mathbf{D}^{(k)}))^2}{|\mathcal{D}|}$.

### 3.2 Per-Edge Weights

We propose a simple mechanism to increase the expressivity of models for data-driven flow routing. As previously mentioned, several works in recent years have begun adopting various graph learning methods for flow routing problems such as variants of Message Passing Neural Networks (Geyer & Carle, 2018; Rusek et al., 2019; Almasan et al., 2021) or Graph Networks (Hope & Yoneki, 2021). In particular, MPNNs derive hidden features $\mathbf{h}_{v_i}^{(l)}$ for node $v_i$ in layer $l+1$ by computing messages $\mathbf{m}^{(l+1)}$ and applying updates of the form:

$$
\begin{aligned}
\mathbf{m}_{v_i}^{(l+1)} &= \sum_{v_j \in \mathcal{N}(v_i)} M^{(l)}\left(\mathbf{h}_{v_i}^{(l)}, \mathbf{h}_{v_j}^{(l)}, \mathbf{x}_{e_{i,j}}\right) \\
\mathbf{h}_{v_i}^{(l+1)} &= U^{(l)}\left(\mathbf{h}_{v_i}^{(l)}, \mathbf{m}_{v_i}^{(l+1)}\right)
\end{aligned}
\tag{2}
$$

where $\mathcal{N}(v_i)$ is the neighborhood of node $v_i$, $\mathbf{x}_{e_{i,j}}$ are features for edge $e_{i,j}$, and $M^{(l)}$ and $U^{(l)}$ are the differentiable message (sometimes also called edge) and vertex update functions in layer $l$. Typically, $M^{(l)}$ is some form of MLP that is applied in parallel when computing the update for each node in the graph. An advantage of applying the same message function $M^{(l)}$ across the entire graph is that the number of parameters remains fixed in the size of the graph, enabling a form of combinatorial generalization (Battaglia et al., 2018). However, while this approach has been very successful in many graph learning tasks such as graph classification, we argue that it is not best suited for flow routing problems.

Instead, for this family of problems, the edges do not have uniform semantics. Each of them plays a different role when the flows are routed over the graph and, as shown in Figure 1, each will take on varying levels of load. Equivalently, from a node-centric perspective, each node should be able to decide flexibly how to distribute several flows of traffic over its neighboring edges. This intuition can be captured by using a different message function $M_{i,j}^{(l)}$ when aggregating messages received along each edge $e_{i,j}$. We call this mechanism *Per-Edge Weights*, or *PEW*. We illustrate the difference between PEW and a typical MPNN in Figure 2.

Let us formulate the PEW architecture by a similar construction to the additive self-attention, across-relation variant of RGAT (Busbridge et al., 2019). Let $\mathcal{N}[v_i]$ and $\mathcal{N}(v_i)$ denote the closed and open neighborhoods of node $v_i$. To compute the coefficients for each edge, one first needs to compute intermediate representations $\mathbf{g}_{v_i,e_{i,j}}^{(l)} = \mathbf{W}_{e_{i,j}}^{(l)} \mathbf{h}_{v_i}$ by multiplying the node features with the per-edge weight matrix $\mathbf{W}_{e_{i,j}}^{(l)}$. Subsequently, the "query" and "key" representations are defined as below, where $\mathbf{Q}_{e_{i,j}}^{(l)}$ and $\mathbf{K}_{e_{i,j}}^{(l)}$ represent per-edge query and key kernels respectively:

$$
\mathbf{q}_{v_i,e_{i,j}}^{(l)} = \mathbf{g}_{v_i,e_{i,j}}^{(l)} \cdot \mathbf{Q}_{e_{i,j}}^{(l)} \text{ and } \mathbf{k}_{v_i,e_{i,j}}^{(l)} = \mathbf{g}_{v_i,e_{i,j}}^{(l)} \cdot \mathbf{K}_{e_{i,j}}^{(l)}.
\tag{3}
$$

Then, the attention coefficients $\zeta_{e_{i,j}}^{(l)}$ are computed according to:

$$
\zeta_{e_{i,j}}^{(l)} = \frac{\exp\left(\text{LeakyReLU}\left(\mathbf{q}_{v_i,e_{i,j}}^{(l)} + \mathbf{k}_{v_j,e_{i,j}}^{(l)} + \mathbf{W}_1^{(l)}\mathbf{x}_{e_{i,j}}\right)\right)}{\sum_{v_k \in \mathcal{N}[v_i]} \exp\left(\text{LeakyReLU}\left(\mathbf{q}_{v_i,e_{i,k}}^{(l)} + \mathbf{k}_{v_k,e_{i,k}}^{(l)} + \mathbf{W}_1^{(l)}\mathbf{x}_{e_{i,k}}\right)\right)},
\tag{4}
$$

Finally, the embeddings are computed as:

$$\mathbf{h}_{v_i}^{(l+1)} = \mathrm{ReLU} \left( \sum_{v_j \in \mathcal{N}(v_i)} \zeta_{e_{i,j}}^{(l)} \mathbf{g}_{v_j, e_{i,j}}^{(l)} \right). \tag{5}$$

We note that the choice of the activation functions and the structure of the weight matrices was made to match the design choices at the basis of the GAT and RGAT architectures. This allows us to conduct a fair assessment the impact of the core difference of PEW with respect to these methods (namely, the per-edge weight parametrizations).

### 3.3 Input features and handling topology variations

The information that is provided to the learning models consists of the demand matrix $\mathbf{D}$, the adjacency matrix $\mathbf{A}$, and the capacities $\kappa$. To use the demands as input features, we consider two formats that appear in prior work, which we term *raw* and *sum*. According to the former, the feature vector $\mathbf{x}_{v_i}^{\mathrm{raw}} \in \mathbb{R}^{2N}$ for node $v_i$ is $[D_{1,i}, \ldots, D_{N,i}, D_{i,1}, \ldots, D_{i,N}]$, which corresponds to the concatenated outgoing and incoming demands, respectively. According to the latter, $\mathbf{x}_{v_i}^{\mathrm{sum}} \in \mathbb{R}^2$ is equal to $[\sum_j D_{i,j}, \sum_i D_{j,i}]$, i.e., it contains the summed (aggregate) demands. For GNN architectures including PEW, the demand vectors are provided as the initial node features $\mathbf{h}_{v_i}^{(0)}$ in the first layer (see Equation 2).

In the context of routing, it is typical to encounter variations in network topology. These variations might include temporary node outages, new nodes joining the network, or certain links becoming unavailable. Handling this with learning architectures is possible, but calls for several adjustments to how the input features are handled. It is required to maintain a mapping from each node to its index in the demand and adjacency matrices, so that the semantics of the flow quantity at a given position in the demand matrix remain consistent (i.e., that the quantity that originates at node $v_i$ is always captured in the $i$-th row).

Therefore, the dimension of the demand matrix $\mathbf{D}$ needs to match the cardinality of the union of the set of nodes that have been present in the network at any point in time. Similarly, if a network expansion is considered and new nodes will be added, one can "reserve" particular indices for these yet-unobserved nodes up to a maximum network size. If a node is not currently active, the corresponding rows and columns in the demand and adjacency matrices are set to 0, to signal the fact that the node is not generating or absorbing any traffic, and that traffic cannot be routed through it. For the GNN models, the absence of a node will be reflected implicitly in the embedding computation, given that no messages will be passed along the edges of the absent node.

We note that the remarks above apply not solely to PEW but all learning architectures for modeling the considered Multi-Commodity Network Flow scenario. The mapping mentioned above between a node $v_i$ and an index $i$, required to assemble all demands for providing model inputs, can also be used in PEW to map an edge $e_{i,j}$ between nodes $v_i, v_j$ to a given parametrization $\mathbf{W}_{e_{i,j}}$. Therefore, PEW does not rely on additional assumptions compared to the standard learning architectures when used for MCNF.

## 4 Evaluation protocol

This section describes the experimental setup we use for our evaluation. We focus on a case study on routing flows in computer networks to demonstrate its effectiveness in real-world scenarios, which can be considered representative of a variety of settings in which we wish to predict characteristics of a routing scheme from an underlying network topology and a set of observed demand matrices. We highlight that our evaluation is performed on a set of real-world topologies and uses both traffic generated by a synthetic model as well as in-situ measurements.

**Model architectures.** We compare PEW with three widely used graph learning architectures: the GAT (Veličković et al., 2018), GCN (Kipf & Welling, 2017), and GraphSAGE (Hamilton et al., 2017). We also compare against a standard MLP architecture made up of fully-connected layers followed by ReLU activations. The features provided as input to the five methods are the same: for the GNN methods, the

node features are the demands $\mathbf{D}$ in accordance with the demand input representations defined in Section 3.3, while the edge features are the capacities $\kappa$, and the adjacency matrix $\mathbf{A}$ governs the message passing. For GCN and GraphSAGE, which do not support edge features, we include the mean edge capacity as a node feature. For the MLP, we unroll and concatenate the demand input representation derived from $\mathbf{D}$, the adjacency matrix $\mathbf{A}$, and all edge capacities $\kappa$ in the input layer. We note that other non-ML baselines, such as Linear Programming, are not directly applicable for this task: while they can be used to derive a routing strategy, in this section the goal is to predict a property of an existing routing strategy (SSP or ECMP, as defined in Section 3.1).

**Network topologies.** For the synthetic traffic evaluation, we consider real-world network topologies that are part of the Repetita and Internet Topology Zoo repositories (Gay et al., 2017; Knight et al., 2011). In case there are multiple snapshots of the same network topology, we only use the most recent so as not to bias the results towards these graphs. We limit the size of the considered topologies to between $[20, 100]$ nodes, which we note is still substantially larger than topologies used for training in prior work on ML for routing flows. Furthermore, we only consider heterogeneous topologies with at least two different link capacities. Given the traffic model above, for some topologies the MLU dependent variable is nearly always identical regardless of the demand matrix, making it trivial to devise a good predictor. Out of the 39 resulting topologies, we filter out those for which the minimum MLU is equal to the 90th percentile MLU over 100 demand matrices, leaving 17 unique topologies. The properties of these topologies are summarized in Appendix D. For the experiments involving topology variations, they are generated as follows: a number of nodes to be removed from the graph is chosen uniformly at random in the range $[1, \frac{N}{5}]$, subject to the constraint that the graph does not become disconnected. Demand matrices are generated starting from this modified topology. For the real-world traffic experiments, we use the 4 topologies from the SNDlib (Orlowski et al., 2010; Idzikowski et al., 2011; Koster et al., 2010) repository that are in this network size range and for which dynamic traffic measurements are available.

**Synthetic traffic generation.** In order to generate synthetic flows of traffic, we use the "gravity" approach proposed by Roughan (2005). Akin to Newton's law of universal gravitation, the traffic $D_{i,j}$ between nodes $v_i$ and $v_j$ is proportional to the amount of traffic, $D_i^{\text{in}}$, that enters the network via $v_i$ and $D_j^{\text{out}}$, the amount that exits the network at $v_j$. The values $D_i^{\text{in}}$ and $D_j^{\text{out}}$ are random variables that are identically and independently distributed according to an exponential distribution. Despite its simplicity in terms of number of parameters, this approach has been shown to synthesize traffic matrices that correspond closely to those observed in real-world networks (Roughan, 2005; Hartert et al., 2015). We additionally apply a rescaling of the volume by the MLU (defined in Section 3.1) under the LP solution of the MCNF formulation, as recommended in the networking literature (Haddadi & Bonaventure, 2013; Gvozdiev et al., 2018).

**Datasets.** We use disjoint datasets $\mathcal{D}^{\text{train}}$, $\mathcal{D}^{\text{validate}}$, $\mathcal{D}^{\text{test}}$ of demand matrices and corresponding MLU values. The ground truth MLU depends on the topology, demand matrix, and routing strategy (SSP or ECMP) and is computed when the available demand matrices are split into training, validation and test sets (further details are provided in the Appendix). Both the demands and capacities are standardized by dividing them by the maximum value across the union of the datasets. The synthetic traffic datasets contain $10^3$ demand matrices each. The real-world traffic measurements from SNDlib are captured at different resolutions (once every 15 min for the Geant, Germany50 and Nobel-Germany topologies and once every 5 minutes for the Abilene network). We use 7 days of data for Abilene and Geant networks and 1 day (the maximum available) for Germany50 and Nobel-Germany, which are split temporally into training, validation, and test sets of equal size. As shown in the Appendix, the synthetic datasets for the smallest topology contain $1.2 * 10^6$ flows, while the datasets for the largest topology consist of $2 * 10^7$ flows. The SNDlib datasets used in the experiments contain between $8.3 * 10^3$ and $9.7 * 10^4$ flows.

**Training and evaluation protocol.** Training and evaluation are performed separately for each graph topology and routing scheme. All methods are given an equal grid search budget of 12 hyperparameter configurations whose values are provided in the Appendix. To compute means and confidence intervals, we repeat training and evaluation across 10 different random seeds. Training is done by mini-batch SGD using the Adam optimizer (Kingma & Ba, 2015) and proceeds for 3000 epochs with a batch size of 16. We perform early stopping if the validation performance does not improve after 1500 epochs, also referred to as "patience" in other graph learning works (Veličković et al., 2018; Errica et al., 2020). Since the absolute value

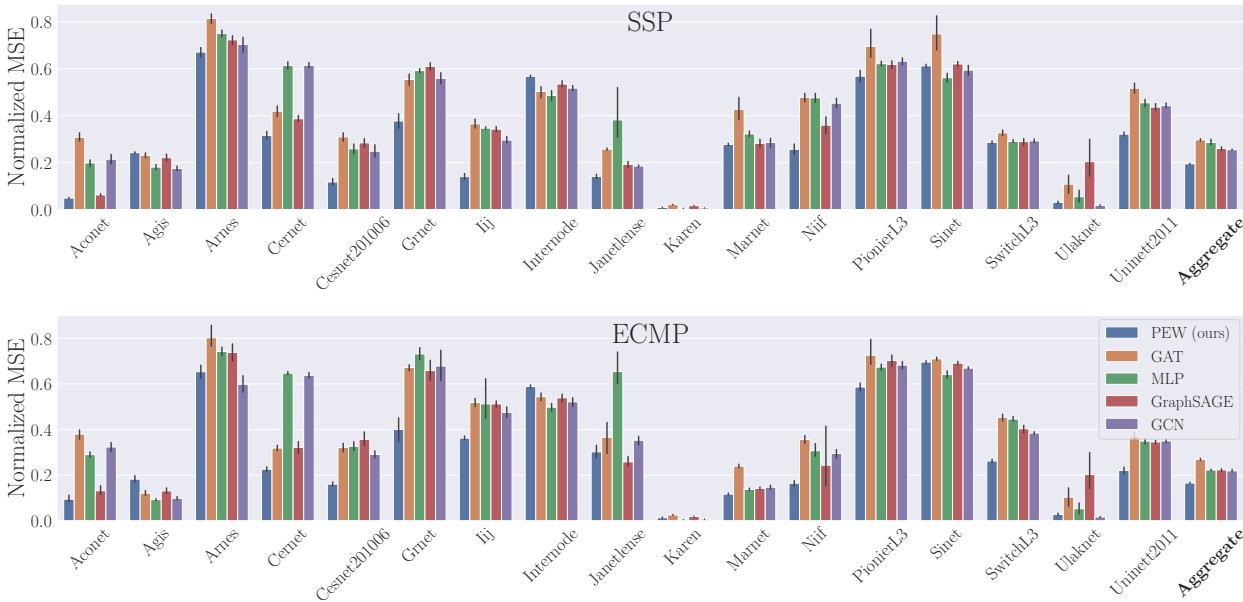

Figure 3: Normalized MSE obtained by the predictors on different topologies with synthetically generated traffic for the SSP (top) and ECMP (bottom) routing schemes. Lower values are better. PEW improves over vanilla GAT substantially and performs best out of all architectures on aggregate. A well-tuned MLP is competitive with the other GNNs. The difficulty of the prediction task on a given topology is consistent across learning architectures.

of the MLUs varies significantly in datapoints generated for different topologies, we apply a normalization when reporting results such that they are comparable. Namely, the MSE of the predictors is normalized by the MSE of a simple baseline that outputs the average MLU for all DMs in the provided dataset. We refer to this as Normalized MSE (NMSE).

**Scale of experiments.** Given the range of considered graph learning architectures, hyperparameter configurations, network topologies and routing models, to the best of our knowledge, our work represents the most extensive suite of benchmarks on graph learning for the MCNF problem to date. The primary experiments consist of 25200 independent model training runs, while the entirety of our experiments comprise 86400 runs. We believe that this systematic experimental procedure and evaluation represents in itself a significant contribution of our work and, akin to (Errica et al., 2020) for graph classification, it can serve as a foundation for members of the graph learning community working on MCNF scenarios to build upon.

## 5 Evaluation results

**Benefits of PEW for flow routing.** The primary synthetic traffic results are shown in Figure 3, in which we compare the normalized MSE obtained by the 5 architectures on the 17 topologies. The two rows correspond to the SSP and ECMP schemes respectively. We find that PEW improves the predictive performance over a vanilla GAT in nearly all (88%) of the settings tested, and that it performs the best out of all predictors in 64.7% of cases. This demonstrates the benefit of the algorithmic alignment between PEW and the routing algorithms, in particular compared to a standard GNN, whose single parametrization per link constrains the embedding computation unnecessarily. Hence, it highlights the importance of parametrizing links differently, suggesting that it provides an effective inductive bias for this family of problems. Interestingly, the MLP performs better than GAT in 80% of the considered cases, and is competitive with GCN and GraphSAGE. This echoes findings in other graph learning works (Errica et al., 2020), i.e., the fact that a well-tuned MLP can be competitive against GNN architectures and even outperform them. Moreover, both

Table 1: Mean Reciprocal Rank and Win Rates for the different predictors, which summarize the performance across multiple topologies. When the graph structure varies by means of different subsets of nodes being present and generating demands (columns marked "Variations"), PEW maintains the overall best performance. The relative performance of the MLP decreases, suggesting that the inductive bias of GNNs is helpful when the underlying graph structure changes.

| $\mathscr{R}$ | Metric | PEW (ours) Original $G$ | Variations | GAT Original $G$ | Variations | MLP Original $G$ | Variations | GraphSAGE Original $G$ | Variations | GCN Original $G$ | Variations |
|---|---|---|---|---|---|---|---|---|---|---|---|
| SSP | MRR ↑ | **0.798** | **0.747** | 0.252 | 0.240 | 0.419 | 0.396 | 0.367 | 0.349 | 0.448 | 0.551 |
|  | WR ↑ | **70.588** | **58.824** | 0.000 | 0.000 | 17.647 | 11.765 | 0.000 | 5.882 | 11.765 | 23.529 |
| ECMP | MRR ↑ | **0.734** | **0.755** | 0.250 | 0.254 | 0.462 | 0.413 | 0.381 | 0.338 | 0.456 | 0.524 |
|  | WR ↑ | **58.824** | **58.824** | 0.000 | 0.000 | 23.529 | 11.765 | 5.882 | 5.882 | 11.765 | 23.529 |

the relative differences between predictors and their absolute normalized MSEs are fairly consistent across the topologies.

**Varying graph structure.** Next, we investigate the impact of variations in topology on predictive performance. In this experiment, the sole difference wrt. the setup described above is that the datasets contain $10^3$ demand matrices that are instead distributed on 25 variations in topology of the original graph (i.e., we have 40 DMs per variation making up each dataset). To evaluate the methods, we use two ranking metrics: the Win Rate (WR) is the percentage of topologies for which the method obtains the lowest NMSE, and the Mean Reciprocal Rank (MRR) is the arithmetic average of the complements of the ranks of the three predictors. For both metrics, higher values are better. Results are shown in Table 1. PEW remains the best architecture and manifests a decrease in MRR for SSP and a gain for ECMP. We also find that the relative performance of the GCN increases while that of the MLP decreases when varying subsets of the nodes in the original graph are present. This suggests that GNN-based approaches are more resilient to changes in graph structure (e.g., nodes joining and leaving the network), a commonly observed phenomenon in practice.

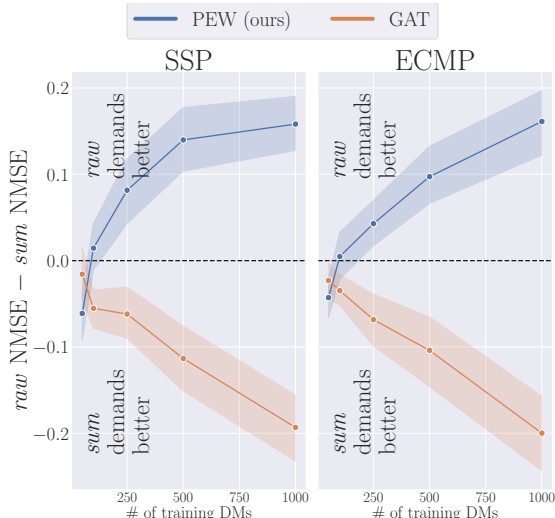

Figure 4: Difference in normalized MSE between the *raw* and *sum* demand input representations as a function of the number of training datapoints for PEW and GAT for the SSP (left) and ECMP routing schemes (right). As the dataset size increases, PEW is able to exploit the granular demand information, while GAT performs better with a lossy aggregation of the demand information.

**Best demand input representation.** To compare the two demand input representations, we additionally train the model architectures on subsets of $5\%, 10\%, 25\%$ and $50\%$ of the synthetic datasets. Recall that the *raw* representation contains the full demand matrix while the *sum* representation is a lossy aggregation of the same information. The latter may nevertheless help to avoid overfitting. Furthermore, given that the distribution of the demands is exponential, the largest flows will dominate the values of the features. Results are shown in Figure 4. The $x$-axis indicates the number of demand matrices used for training and evaluation, while the $y$-axis displays the difference in normalized MSE between the *raw* and *sum* representations, averaged across all topologies. As marked in the figure, $y > 0$ means that the raw representation performs better, while the reverse is true for $y < 0$.

With very few datapoints, the two input representations yield similar errors for both PEW and GAT. Beyond this, two interesting trends emerge: as the number of datapoints increases, PEW performs better with the raw demands, while the vanilla GAT performs better with the lossy representation. This suggests that, while the PEW model is able to exploit the granular information contained in the raw demands, they instead cause the standard GAT to overfit and obtain worse generalization performance.

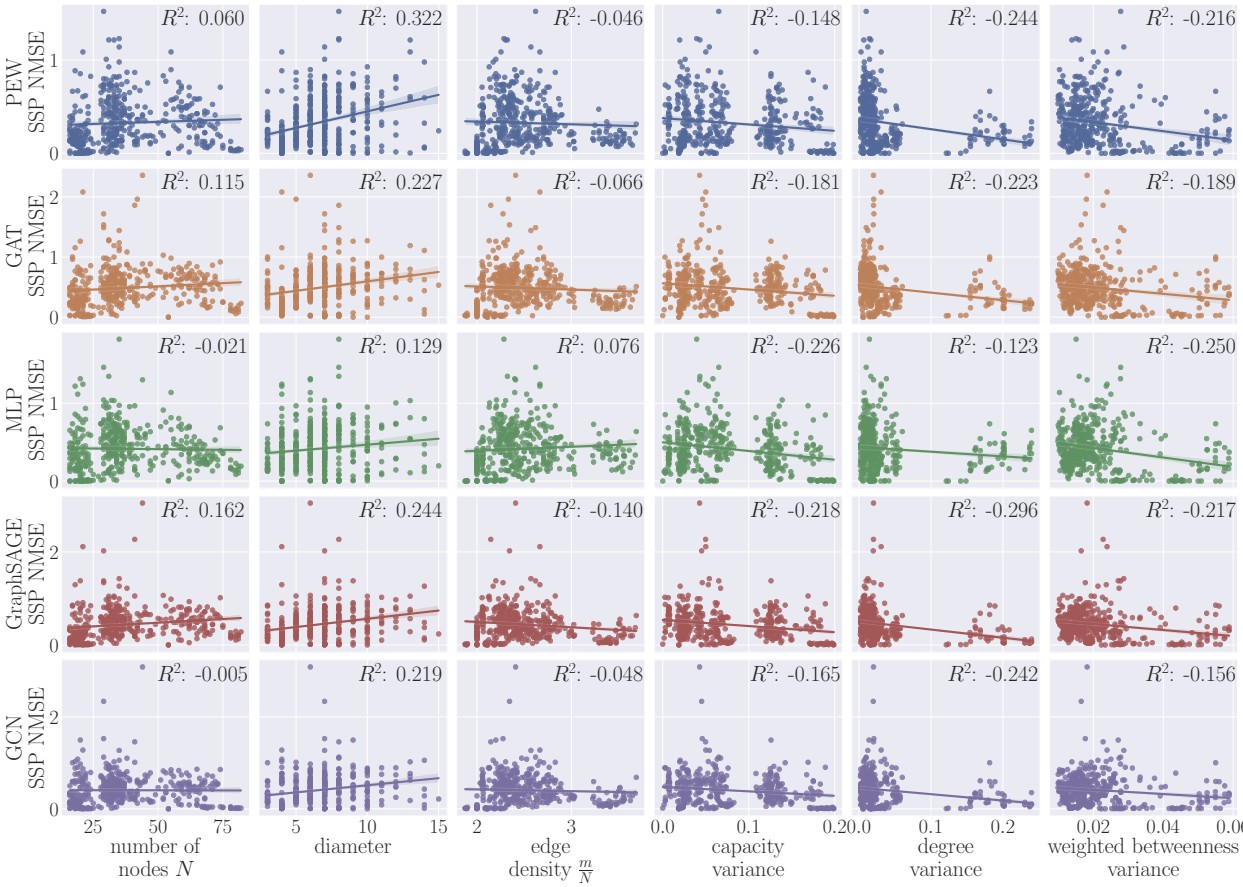

Figure 5: Impact of topological characteristics on the predictive performance of the learning architectures with the SSP routing scheme. Performance degrades as the graph size increases (first 2 columns), but improves with higher levels of heterogeneity of the graph structure (last 3 columns), which is illustrated by the slopes of the regression lines.

**Impact of topology.** This analysis examines the relationship between the topological characteristics of graphs and the relative performance of our proposed model architecture. We leverage the results of the evaluation with variations in topology described above, since we obtain 425 datapoints (17 topologies ∗ 25 variations) for each setting. We fit a linear regression and report the linear $R^2$ correlation coefficient. The six properties that we examine are defined as follows, noting that the first three are global properties while the final three measure the variance in local node and edge properties:

- **Number of nodes**: the cardinality $N$ of the node set $V$;

- **Diameter**: maximum length among pairwise shortest paths;

- **Edge density**: the ratio of links to nodes $\frac{m}{N}$;

- **Capacity variance**: the variance in the normalized capacities $\kappa(e_{i,j})$;

- **Degree variance**: the variance in $\frac{\deg(v_i)}{N}$;

- **Weighted betweenness variance**: the variance in a weighted version of betweenness centrality (Brandes, 2001) measuring the fraction of all-pairs shortest paths passing through each node.

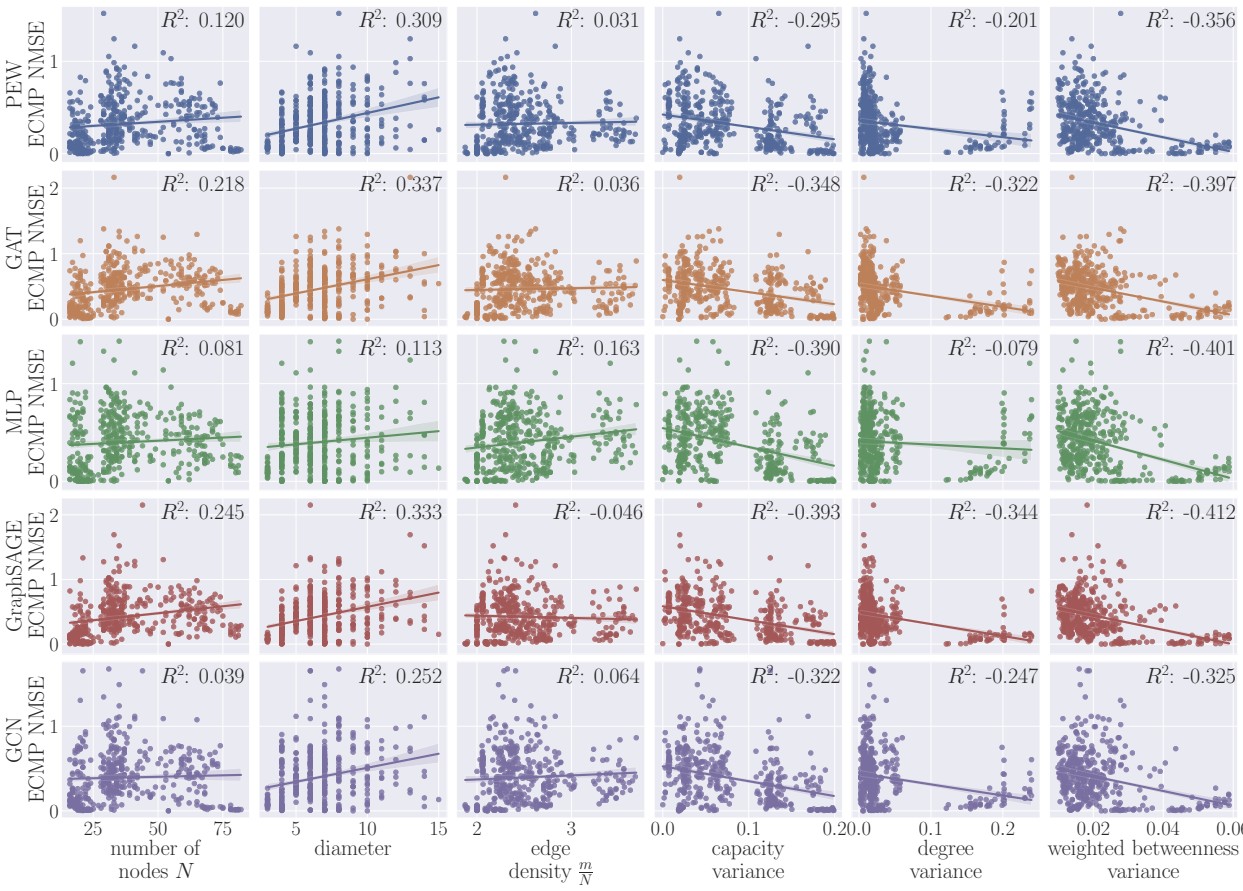

Figure 6: Impact of topological characteristics on the predictive performance of the learning architectures with the ECMP routing scheme. Performance degrades as the graph size increases (first 2 columns), but improves with higher levels of heterogeneity of the graph structure (last 3 columns), which is illustrated by the slopes of the regression lines. The correlations tend to be stronger compared to those obtained with SSP.

The results of this analysis are shown in Figures 5 and 6. As in the previous set of experiments, the normalized MSE of the considered model is shown on the $y$-axis, while the $x$-axis measures properties of the graphs. Each datapoint represents a variation on one of the 17 topologies. We find that topological characteristics do not fully determine model performance. Nevertheless, it is possible to derive a series of observations related to them. Generally, the performance of the method decreases as the size of the graph grows in number of nodes and diameter (metrics that are themselves correlated). This result can be explained by the fact that our experimental protocol relies on a fixed number of demand matrices, which represent a smaller sample of the overall distribution of demand matrices as the graph increases in size. Consequently, this may result in a model exhibiting poorer generalization properties from the training phase to the test phase. On the other hand, the performance of the method typically improves with increasing heterogeneity in terms of node and link-level properties (namely, variance in the capacities and degree / weighted betweenness centralities). The relationship between the NMSE and some of the properties (e.g., weighted betweenness) may be non-linear. The results are fairly consistent among the different learning architectures, suggesting that the difficulty of the prediction task is similar for all the models. This observation is further corroborated by the results in Figure 3, which illustrates that a topology that poses challenges for one learning model also presents difficulties for the others, and vice versa. Lastly, the correlations tend to be stronger for the ECMP routing scheme, which indeed may exhibit more diverse routing paths than SSP and hence be influenced more significantly by topological structures.

Table 2: Results obtained with real-world traffic matrices from the SNDlib dataset. PEW yields the best performance overall out of the considered methods, followed by GraphSAGE. In relative terms, the results for PEW are slightly better than those obtained on synthetic data. The large amount of traffic data available, its regularity, as well as the small topology size explain the low absolute errors for Abilene and Geant.

| $\mathcal{R}$ | Topology | Metric | PEW (ours) | GAT | MLP | GraphSAGE | GCN |
|---|---|---|---|---|---|---|---|
| SSP | Abilene | MSE ↓ | **0.000004** | 0.000018 | 0.000007 | 0.000012 | 0.000024 |
| | Geant | MSE ↓ | **0.000003** | 0.000006 | 0.000008 | 0.000004 | 0.000004 |
| | Germany50 | MSE ↓ | **0.443442** | 0.680482 | 0.501316 | 0.597081 | 0.610358 |
| | Nobel-Germany | MSE ↓ | 0.001134 | 0.000856 | 0.001242 | **0.000819** | 0.001138 |
| ECMP | Abilene | MSE ↓ | **0.000002** | 0.000009 | 0.000004 | 0.000004 | 0.000013 |
| | Geant | MSE ↓ | 0.000002 | 0.000003 | 0.000007 | **0.000002** | 0.000003 |
| | Germany50 | MSE ↓ | **0.257441** | 0.286857 | 0.604823 | 0.407461 | 0.323538 |
| | Nobel-Germany | MSE ↓ | **0.000962** | 0.001025 | 0.001141 | 0.001304 | 0.001173 |
| Aggregate | All | WR ↑ | **75.00000** | 0.000000 | 0.000000 | 25.00000 | 0.000000 |
| | | MRR ↑ | **0.854167** | 0.337500 | 0.329167 | 0.493750 | 0.268750 |

Table 3: Profiling results obtained by benchmarking the considered methods. Generally, MLPs are the most resource-efficient, followed by GraphSAGE and GCN, PEW, and finally GAT, which consumes the most resources. PEW requires less memory and computation over GAT in this setting as its feature vector size is set lower than that of GAT to account for its separate per-link parametrizations. All methods require on the order of milliseconds to obtain a prediction, making them suitable for practical networking scenarios.

| | Memory usage (MB) | | | | | Milliseconds per graph | | | | |
|---|---|---|---|---|---|---|---|---|---|---|
| $G$ | PEW | GAT | MLP | GraphSAGE | GCN | PEW | GAT | MLP | GraphSAGE | GCN |
| Aconet | 594 | 712 | 537 | 543 | 545 | 0.648 | 0.777 | 0.063 | 0.222 | 0.377 |
| Agis | 638 | 876 | 537 | 549 | 552 | 0.970 | 1.237 | 0.068 | 0.331 | 0.453 |
| Arnes | 703 | 1078 | 541 | 558 | 562 | 1.480 | 2.343 | 0.071 | 0.390 | 0.545 |
| Cernet | 688 | 1000 | 543 | 556 | 561 | 1.295 | 1.808 | 0.090 | 0.337 | 0.455 |
| Cesnet201006 | 745 | 1148 | 550 | 569 | 581 | 1.809 | 2.599 | 0.098 | 0.394 | 0.514 |
| Grnet | 713 | 1094 | 542 | 560 | 565 | 1.511 | 1.798 | 0.072 | 0.434 | 0.573 |
| Iij | 704 | 1034 | 542 | 553 | 560 | 1.253 | 1.797 | 0.077 | 0.316 | 0.458 |
| Internode | 806 | 1330 | 561 | 582 | 579 | 2.045 | 3.031 | 0.082 | 0.386 | 0.507 |
| Janetlense | 600 | 738 | 536 | 542 | 543 | 0.650 | 0.837 | 0.070 | 0.252 | 0.325 |
| Karen | 632 | 849 | 537 | 549 | 550 | 1.052 | 1.462 | 0.066 | 0.336 | 0.468 |
| Marnet | 571 | 641 | 536 | 540 | 540 | 0.450 | 0.558 | 0.061 | 0.192 | 0.260 |
| Niif | 685 | 1005 | 542 | 557 | 562 | 1.249 | 1.692 | 0.077 | 0.385 | 0.512 |
| PionierL3 | 773 | 1292 | 541 | 568 | 574 | 1.794 | 2.366 | 0.076 | 0.506 | 0.721 |
| Sinet | 846 | 1421 | 571 | 592 | 590 | 2.305 | 3.925 | 0.129 | 0.565 | 0.719 |
| SwitchL3 | 736 | 1151 | 544 | 562 | 565 | 1.293 | 1.886 | 0.077 | 0.394 | 0.477 |
| Ulaknet | 755 | 1064 | 576 | 587 | 575 | 1.616 | 2.878 | 0.126 | 0.381 | 0.395 |
| Uninett2011 | 1021 | 1988 | 563 | 603 | 614 | 3.393 | 5.945 | 0.124 | 0.695 | 0.913 |

**Real-world traffic results.** The results obtained with the real-world traffic measurements from the SNDlib dataset are shown in Figure 2. PEW yields the lowest MSE in 6 out of 8 cases, with GraphSAGE providing the second-best performance. Overall, the performance of PEW on this data slightly exceeds that obtained with synthetic traffic. The absolute values of the errors are noticeably low for Abilene and Geant, for which substantially more traffic traces are available, suggesting high levels of regularity of the traffic.

**Profiling results.** Given the fact that PEW uses different parametrizations for each link, it is important to consider their impact on speed and memory. Therefore, we profile the total process memory usage in megabytes and the average time in milliseconds required to obtain a prediction for a given topology and demand matrix. This is carried out using the most resource-intensive parameter setting for each model, i.e., with the *raw* demands and the largest feature vector sizes for GNNs / largest hidden layer sizes for the MLP. We measure performance over 100 demand matrices such that the dataset size itself does not substantially impact memory usage. Results are shown in Table 3 and further details about the hardware and libraries used are provided in the Appendix. A clear ranking emerges in both memory and speed, with

the MLP architecture predictably requiring the least amount of memory and computation, followed by the GraphSAGE and GCN architectures. The more expressive PEW and GAT architectures are slower and require more memory. PEW consistently yields better resource usage performance over the closely related GAT architecture, given the fact that the feature vector size $\mathbf{h}_{v_i}$ is set lower for PEW (16 vs. 32 for GAT) to account for its separate per-link parametrizations. All the methods yield predictions in a few milliseconds even with the current reference implementation, which is compatible with the typical requirements of practical networking scenarios.

## 6 Limitations

A possible disadvantage of PEW is that the number of parameters grows linearly with the edge count. However, since the same amount of computations is performed compared to the GAT, there is no increase in runtime. Additionally, given the relatively small scale of ISP backbone networks (up to several hundreds of nodes), we have found that the impact on memory usage was not significant, as illustrated in Figure 3. The largest PEW model, used for the Uninett2011 graph, has approximately $8 * 10^5$ parameters. If required, approaches for reducing the parameter count, such as the basis and block-diagonal decompositions proposed by Schlichtkrull et al. (2018), have already been validated for significantly larger-scale relational graphs. Other routing-specific options that may be investigated in future work could be the "clustering" of the edges depending on the structural roles that they play (such as peripheral or core links) or the use of differently parametrized neighborhoods for the regions of the graph, which may perform well if a significant proportion of the traffic is local. A key assumption behind PEW is that, while there may be some variations in the set of nodes present at any given time, the "backbone" of the network remains similar at different timesteps. This is a suitable assumption for a variety of real-world networks, such as the considered ISP topologies, which are characterized by infrequent upgrades. However, performance may degrade in highly dynamic networks, where the timescale of the structural changes is substantially lower than the time needed in order for systems making use of such a predictive model to adapt.

## 7 Conclusion and future work

In this work, we have addressed the problem of data-driven routing of flows across a graph, which has several applications of practical relevance in areas as diverse as logistics and computer networking. We have proposed Per-Edge Weights (PEW), an effective model architecture for predicting link loads in a network based on historical observations, given a demand matrix and a routing strategy. The novelty of our approach resides in the use of weight parametrizations for aggregating messages that are unique for each edge of the graph. The design of PEW is motivated by the algorithmic alignment with that of routing algorithms. Unlike a standard GNN, which relies on a single message function, PEW captures the intuition that links serve different roles in routing and hence should be parametrized separately. In a rigorous and systematic evaluation comprising 86400 training runs, we have demonstrated that PEW improves predictive performance over standard graph learning and MLP approaches. Furthermore, we have shown that PEW is able to exploit the full demand matrix, unlike the standard GAT, for which a lossy aggregation of features is preferable. Our findings also highlight the importance of topology for data-driven routing. Given the same number of historical observations, performance typically decreases when the graph grows in size, but increases with higher levels of heterogeneity of local properties.

In future work, we aim to evaluate PEW for routing problems with other learning paradigms, i.e., beyond supervised learning. Unsupervised learning may help to avoid the generation of ground-truth labels and still discover meaningful features for capturing network flows. While this paper has focused on learning the properties of *existing* routing protocols, a goal for subsequent work is to investigate learning *new* routing protocols with reinforcement learning. We hope that the learning architecture we have proposed and the broader insights into this problem space we have obtained will represent a solid basis for the discovery of novel and effective routing schemes using machine learning.

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

## A Supplementary material & code and data availability

The supplementary material has been submitted for review and will be made available online in the case of acceptance. It includes source code, the relevant datasets, as well as additional results in the form of learning curves for the main experiments.

**Instructions for running the code.** Please consult the `README.md` file in the `code/` directory of the supplementary material for instructions on how to run the provided code. In the case of acceptance, our implementation will be made publicly available as Docker containers together with instructions that enable reproducing (up to hardware differences) all the results reported in the paper, including tables and figures.

**Data availability.** The network topology data used with synthetic traffic is part of the Repetita suite (Gay et al., 2017) and it is publicly available at `https://github.com/svissicchio/Repetita` without any licensing restrictions. We also use the synthetic traffic generator from (Gvozdiev et al., 2018), available at `https://github.com/ngvozdiev/tm-gen`. The SNDlib topologies and traffic datasets are also available at `https://sndlib.put.poznan.pl/` without restrictions. Scripts are provided to convert the topologies and demand matrices to the suitable input format for our evaluation framework. For convenience, the processed network topologies are made available in the `topologies/` directory of the supplementary material.

## B Implementation and runtime details

**Implementation.** We implement all approaches and baselines in Python using a variety of numerical and scientific computing packages (Hunter, 2007; Hagberg et al., 2008; McKinney, 2011; Paszke et al., 2019; Waskom, 2021). For implementations of the graph learning methods, we make use of PyTorch Geometric (Fey & Lenssen, 2019). Due to the relationship between the RGAT and PEW architectures, we are able to leverage the existing RGAT implementation in this library. For the profiling results reported in Section 5, the core software used is Python 3.9.7, PyTorch 1.11, and PyTorch-Geometric 2.10.

**Computing the MLU values.** MLU values are used as prediction targets for the supervised learning setup considered in this work. They are computed and stored for a given demand matrix, topology, and routing scheme when the training, validation and test sets are created. To calculate the MLU, firstly the shortest paths are computed using uniform edge weights for each pair of source and destination nodes. With SSP, there is a single shortest path along which the entire flow quantity is sent; ECMP requires creating a tree data structure so that the flow quantity can be split among equal-cost paths. The loads for each link are obtained by summing the demands (proportional to the split ratio) of all the paths that traverse them. The link utilizations are equal to the loads divided by the capacities, and the Maximum Link Utilization is the largest value among all the link utilizations for a particular demand matrix. On the implementation side, we have extended the `tm-gen` simulator to support ECMP. This can be consulted in the `packaged_deps/tm-gen/src/ws_to_rc.cc` file in the source code.

**Infrastructure and runtimes.** Experiments were carried out on a cluster of 8 machines, each equipped with 2 Intel Xeon E5-2630 v3 processors and 128GB RAM. On this infrastructure, all the experiments reported in this paper took approximately 37 days to complete. The training and evaluation of models were performed exclusively on CPUs.

## C Hyperparameter details

All methods are given an equal grid search budget of 12 hyperparameter configurations consisting of the two choices of demand input representations, three choices of learning rate $\alpha$, and two choices of model

Table 4: Hyperparameters used.

| | PEW (ours) | GAT | MLP | GraphSAGE | GCN |
|---|---|---|---|---|---|
| Learning rates $\alpha$ | $\{10^{-2}, 5*10^{-3}, 10^{-3}\}$ | $\{10^{-2}, 5*10^{-3}, 10^{-3}\}$ | $\{10^{-2}, 5*10^{-3}, 10^{-3}\}$ | $\{10^{-2}, 5*10^{-3}, 10^{-3}\}$ | $\{10^{-2}, 5*10^{-3}, 10^{-3}\}$ |
| Demand input representations | $\{raw, sum\}$ | $\{raw, sum\}$ | $\{raw, sum\}$ | $\{raw, sum\}$ | $\{raw, sum\}$ |
| Dimension of feature vector $\mathbf{h}$ | $\{4, 16\}$ | $\{8, 32\}$ | n/a | $\{8, 32\}$ | $\{8, 32\}$ |
| First hidden layer size | n/a | n/a | $\{64, 256\}$ *sum* / $\{64, 128\}$ *raw* | n/a | n/a |
| Training epochs | 3000 | 3000 | 3000 | 3000 | 3000 |
| Patience | 1500 | 1500 | 1500 | 1500 | 1500 |

complexity as detailed in Table 4. For the MLP, subsequent hidden layers contain half the units of the first hidden layer. For the GNN-based methods, sum pooling is used to compute a graph-level embedding from the node-level features. Despite potential over-smoothing issues of GNNs in graph classification (e.g., as described in (Chen et al., 2020)), for the flow routing problem, we set the number of layers equal to the diameter so that all traffic entering the network can also exit, including traffic between pairs of points that are the furthest away in the graph.

## D Additional results

Table 5: Properties of the considered Repetita (top) and SNDlib (bottom) topologies.

| Graph | $N$ | $m$ | Diam. | $\frac{m}{N}$ | Flows in $\mathcal{D}$ |
|---|---|---|---|---|---|
| Aconet | 23 | 62 | 4 | 2.70 | 1587000 |
| Agis | 25 | 60 | 7 | 2.40 | 1875000 |
| Arnes | 34 | 92 | 7 | 2.71 | 3468000 |
| Cernet | 41 | 116 | 5 | 2.83 | 5043000 |
| Cesnet201006 | 52 | 126 | 6 | 2.42 | 8112000 |
| Grnet | 37 | 84 | 8 | 2.27 | 4107000 |
| Iij | 37 | 130 | 5 | 3.51 | 4107000 |
| Internode | 66 | 154 | 6 | 2.33 | 13068000 |
| Janetlense | 20 | 68 | 4 | 3.40 | 1200000 |
| Karen | 25 | 56 | 7 | 2.24 | 1875000 |
| Marnet | 20 | 54 | 3 | 2.70 | 1200000 |
| Niif | 36 | 82 | 7 | 2.28 | 3888000 |
| PionierL3 | 38 | 90 | 10 | 2.37 | 4332000 |
| Sinet | 74 | 152 | 7 | 2.05 | 16428000 |
| SwitchL3 | 42 | 126 | 6 | 3.00 | 5292000 |
| Ulaknet | 82 | 164 | 4 | 2.00 | 20172000 |
| Uninett2011 | 69 | 192 | 9 | 2.78 | 14283000 |
| Abilene | 12 | 30 | 5 | 2.50 | 870912 |
| Geant | 22 | 72 | 5 | 3.27 | 975744 |
| Germany50 | 50 | 176 | 9 | 3.52 | 720000 |
| Nobel-Germany | 17 | 52 | 6 | 3.06 | 83232 |

**Topologies used.** High-level statistics about the considered topologies are shown in Table 5.

**Learning curves.** Representative learning curves are available as part of the supplementary material. For their generation, we report the MSE on the held-out validation set of the best-performing hyperparameter combination for each architecture and demand input representation. To smoothen the curves, we apply exponential weighting with an $\alpha_{\text{EW}} = 0.92$. This value is chosen such that a sufficient amount of noise is removed and the overall trends in validation losses can be observed. We also skip the validation losses for the first 5 epochs since their values are on a significantly larger scale and would distort the plots. As large spikes sometimes arise, validation losses are truncated to be at most the value obtained after the first 5 epochs. An interesting trend shown by the learning curves is that the models consistently require more epochs to reach a low validation loss in the ECMP case compared to SSP, reflecting its increased complexity.

