# OpenReview forum: "Graph Neural Modeling of Network Flows"
_TMLR — Rejected by TMLR_

### Review · Reviewer_bcd6 · 2024-03-27

**Summary Of Contributions:**

This paper presents a method and a set of practical evaluations for the network flow modeling problem using graph neural networks. The proposed method (PEW), personalized edge weights, is a simple extension over RGAT that includes edge-specific parameters to allow for edge-specific message functions. The authors perform evaluations over 17 different topologies and 10^3 demand matrices, whose scale is larger than all previous evaluations. The experimental results show the improvements of PEW over baselines, as well as some other insightful findings, such as how the performance changes with the network topology, etc.

**Audience:**

Yes

**Broader Impact Concerns:**

No.

**Claims And Evidence:**

No

**Requested Changes:**

1. Clarify the problem statement. Does this paper aim to generate routing solutions, or analyze how routing solutions work?
2. How can PEW be generalized to changing network topologies?
3. What is the additional computation cost of PEW?
4. Please provide some numerical results regarding Figure 3 and 5.

**Strengths And Weaknesses:**

I am not an expert in network flow modeling, and thus the strengths and weaknesses may not be very specific to the technical field of this paper.

# Strengths:
1. The paper is in general well-written and easy to follow. I can follow the main ideas of this paper in general.
2. The experimental evaluations are extensive. The authors evaluate on 17 different topologies, while existing works only does one or two. The experiments are done over two routing methods, SSP and ECMP, and on networks with more nodes (20-100) compared to existing works. Although I am not familiar with the related works of this paper, the numbers look good and impressive, and that an extensive evaluation is always welcomed in any field related to machine learning.
3. Analysis is done between the topology properties and the prediction performance in Figure 5. I think this would be of interest to researchers in related fields, as this would provide insights on whether, how and why ML-based methods work on this problem, what properties are welcomed, etc.

# Weaknesses
1. The problem setting is slightly confusing. When I read this paper, I thought that the paper tackles the problem of finding the solution of maximum flows, and the introduction reinforces the impression (Paragraph 2 discusses methods to solve for maximum flow problems). However, in Section 3, the problem formulation is to predict the performance of a routing algorithm (as the supervision label is the Maximum Link Utilization). I wonder which (solving the maximum flow problem, or predicting the performance of a routing algorithm) is the problem tackled in this paper. In addition, if the focus is the former (solving the max flow problem), then how can the obtained model be used to derive the solution for the max flow?
2. The proposed PEW does not seem to generalize across different topologies or emerging edges. If my understanding is correct, each edge would maintain a specific weight matrix to determine its message function. Thus, it seems that this matrix cannot be reused for other edges. In practice, when the topology changes (e.g. new edges emerge), the matrix for the new edge should be obtained from scratch. In my opinion, topology changes in real-world networks are quite common, and I think this becomes a drawback of the proposed PEW.
3. Unclear about the additional computation costs. As each edge contains an additional parameter matrix, we expect that the proposed PEW would require more computation than methods like GAT and GCN. The authors did not give any analysis on this point, which can be improved.
4. The presentation of experimental results can be improved. For example, Figure 3 can be improved by providing information such as 'average improvement over baseline', as this is hard to observe with figures. Also, Figure 5 can be improved with some kind of regression analysis to better show the positive/negative correlations (e.g. R^2).
5. Other questions.
    - It would be interesting if Figure 5 can be analyzed on MLP models.
    - In Table 1, it seems that GAT is pretty poor, which seems slightly counter-intuitive, as GAT bears similar ideas with PEW.

---

> ### Author Response · Authors · 2024-04-23
> **Response to Reviewer bcd6**
>
> We thank the reviewer for taking the time to consider our paper. Please find below our responses to the points that were raised.
>
> > 1. Clarify the problem statement. Does this paper aim to generate routing solutions, or analyze how routing solutions work?
>
> We recognize that the early reference in the Introduction to solving maximum flow problems gives the wrong impression regarding the focus of the paper, which had to be corrected. The focus of the paper is the latter: to predict how existing routing solutions work in unseen conditions. This is one of two ways that such a model may be used for networking. The other is to use the model as an underlying learning representation for a method that generates routing solutions. While we do not consider it in this paper, our model is certainly applicable in this scenario as well.
>
> **Changes made**: We have modified the Introduction to make it clear that the problem we aim to treat is predicting the behavior of an existing routing scheme, and that the development of a suitable learning representation is key to data-driven networking applications. We have also moved the discussion about solving maximum flow problems to the “Related work” section to avoid confusion.
>
> > 2. How can PEW be generalized to changing network topologies?
>
> Just to clarify, PEW is already applicable to - and has been evaluated over - changing network topologies. The details on how this is achieved were missing. Essentially, one needs to track all nodes that appear in the network at any one time, and to maintain a mapping from a node to a particular index to the demand and adjacency matrices. If a node is offline, its corresponding demand and adjacency matrix columns and rows are set to 0 to signify that no traffic is being sent or requested. PEW leverages the node-to-index-mappings to maintain separate per-edge parametrizations, relying on the same assumptions as any other learning-based method for MCNF problems.
>
>  The “Varying graph structure” experiments in Section 5 show that the comparative advantage of PEW is maintained in the presence of variations of the graph structure, i.e., when the set of nodes and edges over which the routing takes place differs (equivalently, a subset of the nodes and edges are not present/available at a given time).
>
> **Changes made**: We have added Section 3.3 to extensively discuss how PEW and other learning methods can be adopted to work with variations in topology.
>
> > 3. What is the additional computation cost of PEW?
>
> Thank you for raising this, we fully agree that this should be expanded and quantified. We would like to first make a few observations regarding the time and memory cost compared to GAT:
>
> -	PEW carries out the same number of computations (e.g., matrix multiplications) as its GAT counterpart. Therefore, all other aspects being equal, it requires the same computational time as GAT.
> -	PEW may require more memory in order to store the additional weight matrices. While GAT has a single weight matrix per layer, PEW has as many weight matrices per layer as the number of edges in the graph.
>
> We have configured and ran a series of benchmarks in order to assess the computational and memory costs of PEW. These results are discussed in the “Profiling results” paragraph in Section 5 and illustrated in Table 3, which shows the memory usage in megabytes of the model evaluation process and average milliseconds per graph for the model forward pass. Generally, MLPs are the most resource-efficient, followed by GraphSAGE and GCN, PEW, and finally GAT, which requires the most resources. PEW requires less memory and computation than GAT in this setting as its feature vector size was set lower (16) than that of GAT (32) to account for its separate per-link parametrizations. All methods require on the order of milliseconds to obtain a prediction, making them suitable for deployment in practical networking scenarios.
>
> **Changes made**: addition of new experiment results in Table 3 and relevant commentary in Section 5, as well as relevant details of the experiment conditions (i.e., software versions used) in the Appendix.

---

> > ### Author Response · Authors · 2024-04-23
> > **Response to Reviewer bcd6 (cont.)**
> >
> > > 4. Please provide some numerical results regarding Figure 3 and 5. […] It would be interesting if Figure 5 can be analyzed on MLP models.
> >
> > Figure 3: we fully agree that the average improvement across all topologies should be easier to discern from this figure, thank you for the suggestion.
> >
> > **Changes made**: We have added an additional column to this plot titled “Aggregate” that shows the mean value for the NMSE metric averaged over all the graphs.
> >
> > Figure 5: we have experimented with running a regression for this data, but 17 datapoints appears to be too few to obtain a meaningful result. We realized that the results of the topology variations experiments may be better-suited for this analysis, given that we considered 25 variations per original topology, yielding a 25-fold increase in the number of datapoints. This is sufficient to run a statistical analysis that includes regression lines with confidence intervals and reports the $R^2$ coefficient.
> >
> > Considering the MLP model for Figure 5 as the reviewer suggested yielded similar results to PEW in terms of the topology-predictability relationships. This prompted us to run this analysis for all the models, revealing that the correlations are quite consistent among them. We have therefore made the decision to include all models in this figure and we have separated SSP and ECMP moving them to the new Figures 5 and 6. Interestingly, this new set of comparisons and analyses reveal that correlations tend to be stronger for the ECMP routing scheme.
> >
> > **Changes made**: we have revised the methodology behind the “Impact of topology” analysis in Section 5 to consider the models trained with topology variations. We have added Figure 5 and Figure 6, which contain results for all learning representations and feature linear regressions and correlation coefficients. We also revised the relevant commentary.
> >
> > > 5.2. In Table 1, it seems that GAT is pretty poor, which seems slightly counter-intuitive, as GAT bears similar ideas with PEW.
> >
> > We agree that this may seem counter-intuitive at first. This can be explained by the fact that the GAT is a substantially more expressive model than its other GNN counterparts (i.e., it contains learnable coefficients for how neighbor features should be weighted, while GCN prescribes that this coefficient is equal to the node degree). This greater expressive power allows the model to easily overfit the demand matrices in the training set while not generalizing well to unseen demand matrices due to its wrong inductive bias. Essentially, PEW addresses this key issue of the GAT when applied to flow routing problems.

---

### Review · Reviewer_X6uN · 2024-04-05

**Summary Of Contributions:**

This work proposes a new method of predicting MLU (maximum link utilization) in a flow network. The work has devised a new neural network architecture and showed good performance on a synthetic dataset.

**Audience:**

Yes

**Broader Impact Concerns:**

No concerns.

**Claims And Evidence:**

No

**Requested Changes:**

1. Justify the value of the research with real problems.

2. Fix the writing issues.

**Strengths And Weaknesses:**

Strength:

1. the work has designed a new architecture and assigned learnable a weight vector to each edge. It is reasonable to do so when the graph structure is fixed while flows vary with different demanding matrices.
2. the model shows better performance than other graph neural networks.

Weakness:

The writing of the submission is unclear. In the definition of MLU(D), it seems to be a scalar because of the maximization operator over all edges. MLU(D) is to be predicted from D.  However, Figure 1 seems to suggest that there is a value to predict for each edge.

The introduction states: "a priori knowledge of the full demand matrix is an unrealistic assumption, as loads in real systems continuously change". However, the proposed model does depend on the full knowledge of D.

Since the problem can be solved with a traditional LP solver, I don't quite understand why a learning model is used to make inaccurate predictions. Furthermore, the experiment is conducted on small-scale synthetic data. So there is a big question mark about the usefulness of the proposed method.

---

> ### Author Response · Authors · 2024-04-23
> **Response to Reviewer X6uN**
>
> We thank the reviewer for taking the time to consider our paper. Please find below our responses to the points that were raised.
>
> > The writing of the submission is unclear. In the definition of MLU(D), it seems to be a scalar because of the maximization operator over all edges. MLU(D) is to be predicted from D. However, Figure 1 seems to suggest that there is a value to predict for each edge.
>
> We agree that Figure 1 gave the wrong impression in this sense, thank you for the observation, we have now corrected it.
>
> **Changes made**: We have modified Figure 1 and its caption to make it clear that there is a single maximum value that is being predicted.
>
> > The introduction states: "a priori knowledge of the full demand matrix is an unrealistic assumption, as loads in real systems continuously change". However, the proposed model does depend on the full knowledge of D.
>
> The key distinction is that the learning model is evaluated on unseen data (possibly many demand matrices), whereas LP solutions can only be computed by knowing in advance a single demand matrix (which is also assumed to be static).
>
> > Since the problem can be solved with a traditional LP solver, I don't quite understand why a learning model is used to make inaccurate predictions. […] So there is a big question mark about the usefulness of the proposed method.
>
> The full problem we consider cannot be solved optimally with LP: a routing configuration that is optimal for one demand matrix and is deployed will be sub-optimal under a different demand matrix.
>
> This is one of the reasons why machine learning models have begun to be used for data-driven routing, as we discuss in the “Introduction” and “Related work” sections. The cited works in the networking community that have considered this type of ML model provide evidence that it can be practically used in two ways: (1) to predict how the network will behave in an unseen scenario (as we do in this paper); and (2) as an underlying learning representation for deriving a routing strategy.
>
> The usefulness of the proposed learning representation is as a component of any data-driven routing solution that uses the MCNF problem formulation.
>
> **Changes made**: We have modified the Introduction to make it clear that the problem we aim to treat is predicting the behavior of an existing routing scheme, and that the development of a suitable learning representation is key to data-driven networking applications. We have also moved the discussion about LP solutions to the “Related work” section, as it gave the wrong initial impression about the specific problem we treat in this paper.
>
> > Furthermore, the experiment is conducted on small-scale synthetic data.
>
> Regarding the scale of experiments, we believe that this is a mistaken impression. Our suite of experiments consists of approximately 80000 individual model training runs. We also note that our models are trained on a substantially higher number of (larger) networks than any of the prior work by a significant margin, a claim that can be verified by consulting the papers.
>
> To be specific, [1] uses one topology with 12 nodes and 32 edges. [2] uses a single topology with 11 nodes and 28 edges. To the best of our knowledge, [3] is the only one that treats larger scale networks: 5 topologies of up to 49 nodes, using 100 demand matrices for the largest networks. A recent work [4] trains models with networks that are sampled to have between 5 and 10 nodes. In contrast, we consider 17 topologies of up to 82 nodes, with 1000 demand matrices each.
>
> As part of this revision, we have also extended the evaluation to consider 4 additional topologies in the SNDlib repository that have dynamic traffic traces available. These results are discussed in the “Real-world traffic results” paragraph in Section 5 and Table 2. PEW also proves to be the best learning representation in the performance evaluation with these real-world datasets.
>
>
> [1] Asaf Valadarsky, Michael Schapira, Dafna Shahaf, and Aviv Tamar. Learning to route. In ACM Workshop on Hot Topics in Networks, 2017.
>
> [2] Oliver Hope and Eiko Yoneki. GDDR: GNN-based Data-Driven Routing. In ICDCS, pages 517–527. IEEE, 2021.
>
> [3] Junjie Zhang, Minghao Ye, Zehua Guo, Chen-Yu Yen, and H Jonathan Chao. Cfr-rl: Traffic engineering with reinforcement learning in SDN. IEEE Journal on Selected Areas in Communications, 38(10):2249–2259, 2020.
>
> [4] Ferriol-Galmés, M., Paillisse, J., Suárez-Varela, J., Rusek, K., Xiao, S., Shi, X., ... & Cabellos-Aparicio, A. (2023). RouteNet-Fermi: Network modeling with graph neural networks. IEEE/ACM Transactions on Networking.

---

### Review · Reviewer_2ytC · 2024-04-12

**Summary Of Contributions:**

This paper focuses on improving the performance of solving the multi-commodity network flow (MCNF) routing problem with GNN model. Its contributions can be summarized as follows:
1. It introduces the new per-edge weights (PEW) method to standard GNN and formulates the model for solving MCNP problem. This model can deal with a wide range of network conditions, including changing topologies and fluctuating traffic patterns.
2. It conducts comprehensive experiments to demonstrate that the proposed PEW method surpasses existing GNN methods and MLP in solving MCNF problems and predicting future Maximum Link Utilization (MLU) within the network.
3. It shows a new perspective, examining the relationship between topology characteristics, such as graph size and node/edge heterogeneity, and the prediction performance of the learning model. The findings suggest that topology has a stronger prediction ability in smaller topologies and those with greater heterogeneity.

**Audience:**

Yes

**Broader Impact Concerns:**

N.A.

**Claims And Evidence:**

Yes

**Requested Changes:**

1. Authors may need to clearly explain the motivation behind predicting MLU over future traffic, and its relation to its routing deployment plan. Authors need to explain the connection between MLU prediction and routing strategies (refer to Weakness 1 for details).
2. Authors may need to provide high-level insights into why and how PEW improves MLU or routing plans. In the current paper, the effectiveness is only demonstrated through experiments; consider adding more explanatory content to motivate this method better.
3. Authors may need to clarify how the input/output format of the learning model adapts to various topology/traffic inputs and how PEW in topologies changes when topologies are changed (refer to weakness 2 for details).
4. Authors may specify the process for generating the ground truth label (future MLU) in the supervised learning model (refer to weakness 3 for details), as this is crucial for model training.
5. Authors may explore extending PEW to other learning methods, such as unsupervised learning, to demonstrate its generality. Unsupervised learning can also save time in preparing ground truth labels, which may require future traffic information that may be obtained with difficulty.
6. Authors may use more realistic traffic datasets, such as those collected from production (e.g., SNDLib), to make their results more convincing. While synthetic traffic with a gravity model in current work approximates real-world traffic, it may be easier for the learning model to learn due to its mathematical formulation nature of gravity model.
7. Authors may compare PEW with more advanced traditional and learning-based methods (refer to weakness 4). Consider comparing with schemes like Smore (NSDI'18), which specifies multiple candidate paths and arbitrary traffic fraction of a path (that is obviously superior to SSP and ECMP),  and DRL-based solutions that focus on topology and traffic changes.

**Strengths And Weaknesses:**

Strengths
1. This paper proposes the PEW methods in GNN models to improve its message aggregating ability from neighboring nodes, which is an aspect that has not been explored before.
2. The PEW method shows strong ability in solving MCNF routing problem in changing topology/traffic scenario, and it can outperform the standard GNN architecture.
3. This paper shows the relationship between topology characteristics and the prediction performance of learning model, which is a novel and interesting aspect that rare papers have explored before.

Weaknesses
1. The paper's motivation requires more clarity. The authors use learning models to directly predict future MLU. However, it's unclear how this approach differs from predicting future traffic and then solving the LP problem to obtain the MLU and routing plans, which is the method commonly used in Traffic Engineering (TE) works. Furthermore, while the authors opt to predict MLU directly, they do not provide design details about the corresponding routing plan associated with this MLU.
2. There is a lack of explanation about the design of the model structure and its adaptability to changing topologies. The authors do not clearly describe the format of the model's input/output, which is essential for understanding how the model handles changing topologies that may result in different input/output formats. Additionally, the authors should explain how the PEW method adjusts when nodes are added or removed from the topologies and whether the topologies need to be re-trained in these scenarios.
3. There is a lack of detail about the generation of ground truth labels. The authors use supervised learning to predict future MLU, but they do not explicitly explain how they generate the ground truth MLU for future traffic. The authors need to specify the methods and details used to generate it. For instance, the authors may use the ground truth traffic in the future and formulate an LP to solve for the MLU. In such cases, the authors should provide clear explanations about how to obtain the ground truth traffic in the future and how to formulate the MLU.
4. The comparison methods used are limited. The authors only compare the PEW method with simple traditional SSP and ECMP methods, as well as learning-based GNN and MLP methods. However, there are more advanced traditional TE methods, such as MPLS, and more advanced learning-based methods, such as deep RL that also aim to handle changing topologies and traffic. The authors should consider comparing their method with these more advanced techniques.

---

> ### Author Response · Authors · 2024-04-23
> **Response to Reviewer 2ytC**
>
> We thank the reviewer for taking the time to consider our paper. Please find below our responses to the points that were raised.
>
> > 1. Authors may need to clearly explain the motivation behind predicting MLU over future traffic, and its relation to its routing deployment plan. Authors need to explain the connection between MLU prediction and routing strategies (refer to Weakness 1 for details).
>
> Works that treat routing problems with machine learning consider either (1) predicting the behavior of an existing routing scheme in unseen circumstances and (2) learning the routing scheme itself. Our work falls in the first category and does not consider explicitly learning a routing strategy. The prediction of MLU over future traffic is performed as a means of benchmarking the suitability of a variety of learning representations for flow routing tasks. Indeed, the proposed method may also be employed as part of a solution for learning the routing strategy itself, but this is not investigated in the current paper.
>
> **Changes made**: We have revised the “Introduction” section to clarify that the task considered in the paper is to predict the behaviour of an existing routing scheme.
>
> > 2. Authors may need to provide high-level insights into why and how PEW improves MLU or routing plans. In the current paper, the effectiveness is only demonstrated through experiments; consider adding more explanatory content to motivate this method better.
>
> The core argument, which is made in the Introduction and Section 3.2 is that PEW provides better algorithmic alignment between the computations executed by the routing algorithm and those executed by the GNN. Namely, for routing flows, each link participates in the routing differently and has its own “state” depending on the flows that traverse it and hence is given a different parametrization in PEW. The typical GNN imposes a single parametrization per link, which constrains the embedding computation unnecessarily.
>
> **Changes made**: we have added an internal reference to the motivation behind the method in the Evaluation Results and Conclusion sections after the experimental results are discussed.
>
> > 3. Authors may need to clarify how the input/output format of the learning model adapts to various topology/traffic inputs and how PEW in topologies changes when topologies are changed (refer to weakness 2 for details).
>
> We agree that this was not discussed in sufficient detail. First of all, one needs to track all nodes that appear in the network at any one time, and to maintain a mapping from a node to a particular index in the demand and adjacency matrices. If a node is offline, its corresponding demand and adjacency matrix columns and rows are set to 0 to signify that no traffic is being sent or requested. PEW leverages the node-to-index-mappings to maintain separate per-edge parametrizations, requiring essentially the same assumptions as any other learning-based method for MCNF problems.
>
> **Changes made**: we have added a new section (3.3) to cover the learning model inputs and how variations in topology are handled (not only by PEW, but by all the predictors that are applied in this scenario).

---

> > ### Author Response · Authors · 2024-04-23
> > **Response to Reviewer 2ytC (cont.)**
> >
> > > 4. Authors may specify the process for generating the ground truth label (future MLU) in the supervised learning model (refer to weakness 3 for details), as this is crucial for model training.
> >
> > We agree that this was not explained in sufficient detail. The ground truth MLU depends on the demand matrix and routing strategy (SSP or ECMP). It is computed at the “dataset generation stage” when the training, validation, and test sets are created (i.e., before the models are trained). The MLU is stored together with the demand matrix, adjacency matrix and link capacities as one “datapoint”.
> >
> > To calculate the MLU, firstly the shortest paths are computed for each pair of source and destination nodes using uniform edge weights. With SSP, there is a single shortest path along which the entire flow quantity is sent; ECMP requires creating a tree data structure so that the flow quantity can be split among equal-cost paths. The loads for each link are obtained by summing the demands (proportional to the split ratio) of all the paths that traverse them. The link utilizations are equal to the loads divided by the capacities. The Maximum Link Utilization is the largest value among all the link utilizations for a particular demand matrix. As far as the implementation is concerned, we have extended the tm-gen simulator to support ECMP. This can be consulted in the `packaged_deps/tm-gen/src/ws_to_rc.cc` file in the source code.
> >
> > As also specified in the response to point 1 above, we do not aim to derive a routing strategy, but to predict the resulting behavior of a known routing strategy in an unseen scenario given a traffic demand matrix and a network topology. We therefore do not use the model to predict future demands in this setup (with respect to which an LP solution is then computed), although we are aware that this is a common way of addressing the TE problem.
> >
> > **Changes made**: we have added the details regarding the generation of the ground truth labels to Section 4 (Datasets paragraph) and the Appendix.
> >
> > > 5. Authors may explore extending PEW to other learning methods, such as unsupervised learning, to demonstrate its generality. Unsupervised learning can also save time in preparing ground truth labels, which may require future traffic information that may be obtained with difficulty.
> >
> > We would like to note that PEW is compatible as-is with other machine learning paradigms (e.g., unsupervised learning and reinforcement learning) and can be applied to any MCNF-compatible learning scenario. Given the positive evidence with supervised learning, we have reason to believe that it would also perform well in the context of other paradigms (similarly to how computer vision architectures that were developed in a supervised learning context have been used successfully as a function approximator in reinforcement learning).
> >
> > We do agree that evaluating the method on another learning paradigm is worthwhile and would further support its generality. However, realistically speaking, adding another learning paradigm altogether requires substantially more time and resources than what can be achieved during a revision window; and we have focused our efforts on extending the supervised learning evaluation with real-world data (see point 6 below). In our opinion, unsupervised learning may be applied to (1) the problem of the prediction of an entry of a demand matrix given the other entries; and (2) the problem of learning a set of features that can faithfully reconstruct the input topology and demands in an auto-encoder style setup.
> >
> > **Changes made**: we have added a discussion on evaluating PEW in unsupervised and reinforcement learning scenarios in the Conclusion section (renamed “Conclusion and future work”).
> >
> > > 6. Authors may use more realistic traffic datasets, such as those collected from production (e.g., SNDLib), to make their results more convincing. While synthetic traffic with a gravity model in current work approximates real-world traffic, it may be easier for the learning model to learn due to its mathematical formulation nature of gravity model.
> >
> > We agree that an evaluation with real-world traffic would strongly support the findings of our work, given a possible alternative explanation for the performance of PEW is an ability to approximate the gravity model.
> >
> > **Changes made**: we have extended our evaluation to consider the 4 networks in the SNDlib repository in the 20-100 node range that have dynamic traffic traces available. These results are discussed in the “Real-world traffic results” paragraph in Section 5 and Table 2. They show that PEW yields the best performance overall out of the considered methods, followed by GraphSAGE. In relative terms, the results for PEW are slightly better than those obtained on synthetic data. On the Abilene and Geant networks, which have a large number of traces available, all models obtain low absolute errors.

---

> > > ### Author Response · Authors · 2024-04-23
> > > **Response to Reviewer 2ytC (cont.)**
> > >
> > > > 7. Authors may compare PEW with more advanced traditional and learning-based methods (refer to weakness 4). Consider comparing with schemes like Smore (NSDI'18), which specifies multiple candidate paths and arbitrary traffic fraction of a path (that is obviously superior to SSP and ECMP), and DRL-based solutions that focus on topology and traffic changes.
> > >
> > > Given that our goal is not to learn the routing strategy itself, our method is not in competition with traditional and learning-based traffic engineering methods. Instead, PEW is compatible with any learning-based solution that uses a MCNF formulation and can be used as a function approximator within it in place of a standard GNN or MLP. Pairing PEW with a DRL solution is an aspect we are very excited about considering in our future work and, as we have also argued in the response to point 5, there are reasons to believe this would lead to gains over approaches that use standard learning representations.
> > >
> > > **Changes made**: we have added a discussion on evaluating PEW in unsupervised and reinforcement learning scenarios in the Conclusion section (renamed “Conclusion and future work”).

---

### Review · Reviewer_wMTy · 2024-04-13

**Summary Of Contributions:**

This paper studies a problem of designing neural networks on graphs to predict the utilization of the network load. The set up is like this:

- At training time, the input consists of a fixed graph (possibly directed, and weighted), plus a collection of traffic demand matrices; In each demand matrix, there is a flow request from one node to a different node.

- Based on this demand matrix and a prespeficied routing scheme (the paper considers two routing schemes, including the shortest path routing, and the ECMP (p.s. it would be good to give the full name of this...) protocol (which splits outgoing traffic among all neighbors on shortest paths to destination if multiple such neighbors exist), one could compute the load of each edge in the graph.

- To evaluate the outcome of this routing scheme, the paper evaluates the Maximum Link Utilization (MLU), which is the max ratio between the load the the capacity. Thus, there is one MLU for each demand matrix.

- The problem is that given a fixed graph, and a collection of (demand matrix, MLU) pairs, is there a way to predict MLU based on the demand matrix.

The paper then went on to design a Multi-Layer Perceptron type method, which introduces an embedding for each node / edge (it would be good to clarify because I'm not completely sure if one or both are used here), and then propose to aggregate the embeddings through the graph edges. This is called the Per-Edge Weight (PEW) method.

The paper then compares PEW with a few standard neural networks, including GAT (graph attention), GCN, and GraphSAGE.

- The main claim is that PEW can generally outperform these alternative approaches.

- The data sets (namely, the traffic flows) are generated according to synthetic data, but the graphs are inherited from (17) real world Internet Service Provider topologies.

- The paper also discusses the impact of varied graph structure and how it affects the prediction outcomes.

**Audience:**

Yes

**Claims And Evidence:**

Yes

**Requested Changes:**

1) There are a number of design choices in the PEW method, such as the activation function (LeakyReLU vs ReLU), dimension of the weight matrices, that are not very clearly explained. I feel like they could be better justified, like providing some ablation studies to justify their choices.

2) The performance of GNNs is very sensitive to the hyper-parameter choices of the training algorithms, and the number of layers built in. Did the authors explore these parameter choices in their experiments? I feel like it would be good to have a robustness check, at least I would want to make sure, since this might affect the conclusion of this paper.

3) I feel like the results of this paper could benefit from having more explanations---is there a compelling reason for why PEW outperforms the other baselines? What would be a justification / post hoc explanation for the empirical findings?

4) The paper is relatively easy to follow for me, but I do encounter hiccups while reading (note that I only read through the main text but I didn't carefully check the appendix, which seems to contain a lot of similar looking figures, that I feel requires some clean up as well).

In particular, see my comments inlined with my general summary above. I feel the paper could benefit from some polish in its writing. For instance, the captions of the figures don't really give me an easily accessible summary of what's going on inside the figure. These aspects limit the accessibility of the current manuscript.

**Strengths And Weaknesses:**

The paper makes a nice connection between a classical combinatorial problem of network flow to a modern setup of learning on graphs. I'm not super familiar with the literature in this line but I do appreciate the work of the authors to establish this connection. It is an interesting observation that neural networks (if at all possible) can predict the outcomes of the shortest path flow schemes up to a reasonable accuracy!

With regarding to the limitation of existing Graph Neural Network (GNN) architectures on solving combinatorial optimization problems, there's a recent article which offers some theoretical perspectives into this:

Gamarnik, David. "Barriers for the performance of graph neural networks (GNN) in discrete random structures." Proceedings of the National Academy of Sciences 120.46 (2023): e2314092120.

I also enjoyed the discussion on "Algorithmic Reasoning" and how it may relate to this paper.

---

> ### Author Response · Authors · 2024-04-23
> **Response to Reviewer wMTy**
>
> We thank the reviewer for taking the time to consider our paper. Please find below our responses to the points that were raised.
>
> > With regarding to the limitation of existing Graph Neural Network (GNN) architectures on solving combinatorial optimization problem, there's a recent article […]
>
> **Changes made**: Many thanks for the pointer, we have added this reference as well as other prior works on GNNs/ML for combinatorial optimization to the “Related work” section.
>
> > 1. There are a number of design choices in the PEW method, such as the activation function (LeakyReLU vs ReLU) […] that are not very clearly explained.
>
> The choice of the LeakyReLU activation function and the structure of the weight matrices was made to match the GAT and RGAT architectures in order allow for a fair comparison with them. In fact, this enables us to assess the impact of the core difference of PEW (the per-edge weight parametrizations) while keeping the other design choices the same. We agree that exploring these aspects of the design space further may lead to better performance, but it was not our main goal.
>
> **Changes made**: We have modified Section 3.2 to provide this justification.
>
> > 2. The performance of GNNs is very sensitive to the hyper-parameter choices of the training algorithms, and the number of layers built in. Did the authors explore these parameter choices in their experiments? I feel like it would be good to have a robustness check, at least I would want to make sure, since this might affect the conclusion of this paper.
>
> The 5 learning architectures that are compared in our paper were tuned on a held-out validation set and each method was given an equal budget of 12 possible hyperparameter choices. This is discussed in Section 4 and the specific hyperparameter values are provided in the Appendix. Therefore, we consider that the results are robust with respect to the hyperparameter choices. We agree, in general, that the performance of GNNs is sensitive to the values of the hyperparameters, which is why we have followed a rigorous experimental procedure.
>
> > 3. I feel like the results of this paper could benefit from having more explanations---is there a compelling reason for why PEW outperforms the other baselines? What would be a justification / post hoc explanation for the empirical findings?
>
> The core argument, which is made in Sections 1 and 3.2 but was not discussed afterwards, is that PEW provides better algorithmic alignment between the computations executed by the routing algorithm and those executed by the GNN. Namely, for routing flows, each link participates differently and has its own “state” depending on the flows that traverse it. For this reason, it is given a different parametrization in PEW. The typical GNN imposes a single parametrization per link which, in our opinion, constrains the embedding computation unnecessarily.
>
> **Changes made**: We have included this explanation in Section 5 (see discussion of the results) as well as in the Conclusion. We note that this is not merely a post-hoc explanation but was our original motivation in devising the architecture in the first place.
>
> > 4. The paper is relatively easy to follow for me, but I do encounter hiccups while reading (note that I only read through the main text but I didn't carefully check the appendix, which seems to contain a lot of similar looking figures, that I feel requires some clean up as well).
>
> **Changes made**: We have removed the learning curves from the Appendix and placed them in the supplementary material, as we agree that they make the manuscript substantially lengthier without adding substantial information.
>
> > In particular, see my comments inlined with my general summary above. I feel the paper could benefit from some polish in its writing. For instance, the captions of the figures don't really give me an easily accessible summary of what's going on inside the figure. These aspects limit the accessibility of the current manuscript.
>
> **Changes made**: We have revised the captions of the figures and tables to be as detailed and self-explanatory as possible - many thanks for this suggestion. We have also polished the writing in several parts of the manuscript.

---

### Author Response · Authors · 2024-04-23
**General note to the action editor and all reviewers**

We would like to sincerely thank the action editor and all the reviewers for considering our paper. We prepared a revised version to address all the comments, which have contributed to substantially improving the quality of the manuscript. We have uploaded a new revision of the paper, with the additions and changes highlighted in light blue.

For global visibility, we have listed the core changes below, and additionally we have replied individually to each reviewer. The core changes are:

-	We have carried out an evaluation with real-world traffic traces over 4 topologies from the SNDlib dataset. These results, presented in Table 2 and discussed in Section 5, show that the proposed PEW method obtains good performance with real traffic measurements and is not superior solely in scenarios with traffic generated by the synthetic gravity model. We have also modified Section 4 to provide the relevant details for these experiments.
-	We have profiled the memory usage and computational time of the methods in Table 3, showing that our method does not incur substantial resource costs, and it is more efficient (with the considered parameters) than the GAT.
-	We have revised Figure 5 and Figure 6 using data from the experiments with topology variations instead of those with a single static network; the number of datapoints allowed us to run regressions and compute a linear correlation coefficient to support the “exploratory” findings that we presented in the previous version of the manuscript. The plots in these Figures now include data from all the learning architectures. They show that the performance is consistent across all of them.
-	We have added a new section (3.3) that provides clarifications on how the input features are processed and how topology variations are handled.
-	Additional modifications were made throughout the paper (discussed in the relevant responses) to improve its clarity and readability.

We are open to clarifying any further points during the remainder of the discussion period.


Your sincerely,

The Authors

---

### Comment · Action_Editor_y19C · 2024-05-14
**Reject (Encouraging Resubmission)**

The paper deals with a relevant and timely problem of devising GNN approaches for network routing. The paper builds upon the very simple idea of predicting edge weights for flow distributions. The reviews are rather mixed, with all reviewers pointing out some valid positives and some valid negatives. Unfortunately, my major concern - shared by one of the Reviewers - is that despite pointing out the relevance of their solution when demand matrices are not known a priori,  the authors seem to use the demand matrix as part of their architecture.

In one of the first papers on the subject, Learning how to route, it is made clear that one can only use historical data about network demands and then hope that the demands from the past have some bearing on the demands in the future. At least to me, it is not clear (given the current writeup) that the authors did not make use of a "current" demand matrix. Maybe the writing was confusing, or both me and the reviewer misunderstood something. But in both cases, the authors really need to make their modeling and data availability assumptions much more transparent. Furthermore, I also agree with one of the Reviewers that pointed out that "Importantly, the true labels of MLUs are computed from LP solutions" - if that is the case (and the authors should make this point much more transparent), than the authors should provide more in-depth clarifications.

In summary, the issue with the paper is that it is confusing to the readers in terms of the modeling and demand matrix availability assumptions. Until this problem has been settled I cannot recommend the paper for acceptance.

---

### Decision · Action_Editor_y19C · 2024-05-14

**Recommendation:** Reject

**Comment:**

The reviews are rather mixed, with all reviewers pointing out some valid positives and some valid negatives. Unfortunately, my major concern - shared by one of the Reviewers - is that despite pointing out the relevance of their solution when demand matrices are not known a priori, the authors seem to use the demand matrix as part of their architecture.

In one of the first papers on the subject, Learning how to route, it is made clear that one can only use historical data about network demands and then hope that the demands from the past have some bearing on the demands in the future. At least to me, it is not clear (given the current writeup) that the authors did not make use of a "current" demand matrix. Maybe the writing was confusing, or both me and the reviewer misunderstood something. But in both cases, the authors really need to make their modeling and data availability assumptions much more transparent. Furthermore, I also agree with one of the Reviewers that pointed out that "Importantly, the true labels of MLUs are computed from LP solutions" - if that is the case (and the authors should make this point much more transparent), than the authors should provide more in-depth clarifications.

In summary, the issue with the paper is that it is confusing to the readers in terms of the modeling and demand matrix availability assumptions. Until this problem has been settled I cannot recommend the paper for acceptance.

**Audience:**

As already pointed out, the paper addresses a very important and interesting problem. A substantial TMLR audience may be interested in the subject matter and practitioners in the field of network analytics may find it valuable as well.

**Claims And Evidence:**

The paper deals with a relevant and timely problem of devising GNN approaches for network routing. The paper builds upon the very simple idea of predicting edge weights for flow distributions and presents extensive results on several datasets. The idea itself is very straightforward but the modeling assumptions are unclear and hence do not make a convincing case that this method can truly address unknown demand matrix scenarios.

**Resubmission Of Major Revision:**

The authors may consider submitting a major revision at a later time.